# Identifying a Common Functional Framework for Apathy Large-Scale Brain Network

**DOI:** 10.3390/jpm11070679

**Published:** 2021-07-19

**Authors:** Vincenzo Alfano, Mariachiara Longarzo, Giulia Mele, Marcello Esposito, Marco Aiello, Marco Salvatore, Dario Grossi, Carlo Cavaliere

**Affiliations:** 1IRCCS SDN, Via Emanuele Gianturco, 113, 80142 Naples, Italy; vincenzo.alfano@synlab.it (V.A.); mariachiara.longarzo@synlab.it (M.L.); marco.aiello@synlab.it (M.A.); direzionescientifica.irccssdn@synlab.it (M.S.); carlo.cavaliere@synlab.it (C.C.); 2AOU Cardarelli, 80131 Naples, Italy; marcelloesposito@live.it; 3Department of Psychology, Università degli Studi della Campania Luigi Vanvitelli, 81100 Caserta, Italy; dario.grossi@unicampania.it

**Keywords:** apathy, fMRI, frontotemporal dementia, Parkinson’s disease, neuropsychology

## Abstract

Apathy is a neuropsychiatric condition characterized by reduced motivation, initiative, and interest in daily life activities, and it is commonly reported in several neurodegenerative disorders. The study aims to investigate large-scale brain networks involved in apathy syndrome in patients with frontotemporal dementia (FTD) and Parkinson’s disease (PD) compared to a group of healthy controls (HC). The study sample includes a total of 60 subjects: 20 apathetic FTD and PD patients, 20 non apathetic FTD and PD patients, and 20 HC matched for age. Two disease-specific apathy-evaluation scales were used to measure the presence of apathy in FTD and PD patients; in the same day, a 3T brain magnetic resonance imaging (MRI) with structural and resting-state functional (fMRI) sequences was acquired. Differences in functional connectivity (FC) were assessed between apathetic and non-apathetic patients with and without primary clinical diagnosis revealed, using a whole-brain, seed-to-seed approach. A significant hypoconnectivity between apathetic patients (both FTD and PD) and HC was detected between left planum polare and both right pre- or post-central gyrus. Finally, to investigate whether such neural alterations were due to the underlying neurodegenerative pathology, we replicated the analysis by considering two independent patients’ samples (i.e., non-apathetic PD and FTD). In these groups, functional differences were no longer detected. These alterations may subtend the involvement of neural pathways implicated in a specific reduction of information/elaboration processing and motor outcome in apathetic patients.

## 1. Introduction

Apathy is a neuropsychiatric condition characterized by reduced motivation, initiative, and interest in daily life activities. According to Levy and Dubois, apathetic syndrome is articulated in three domains [1]: cognitive apathy is defined by difficulty in elaborating action plans and by a loss of goal-directed cognitive processing; behavioral apathy is characterized by difficulties in activating independent thoughts and actions; and emotional apathy corresponds to difficulties in establishing links between emotional stimuli and proper behavior.

Apathy is commonly reported in several neurodegenerative disorders, such as Alzheimer’s disease (AD) [2], mild cognitive impairment (MCI), and Parkinson’s disease (PD) [3]. Apathy is severely present in frontotemporal dementia (FTD) [4] and commonly belongs to behavioral abnormalities in behavioral variant of FTD (bvFTD).

The great majority of available neuroimaging studies about apathy specifically in FTD and PD consider structural and metabolic changes. Alterations in cortical thickness, white matter, perfusion, and metabolism have been frequently reported [5,6,7,8,9]. Overall, structural and metabolic changes in patients with neurodegenerative disorders and apathy mainly affect left inferior frontal gyrus (part of central executive network) activated by stimulus-driven cognitive and affective processing [5].

In FTD patients, apathy is reported to be associated with atrophy of the frontal lobes (including lateral, orbital, and medial regions) and with right frontal cortex hypoperfusion [6]. Eslinger et al. [6] suggested that parallel decline in motivational and executive functions and social domains in bvFTD were associated with a large-scale neural network that includes basal ganglia and prefrontal regions. In PD patients, apathy represents principal non-motor symptoms. Several MRI studies showed a reduction of gray matter thickness of the inferior frontal cortex, premotor, and posterior cingulate cortices [10]. Additionally, in PD patients, apathy may be associated with brain hypometabolism in frontal cortex, limbic lobe, and cerebellum negatively correlated to apathy severity [11].

Studies about the possible relationship between apathy on functional connectivity (FC) are rarer. Farb and colleagues [12] showed that in bvFTD, an increased prefrontal cortex connectivity was associated with apathy, which contributes to dementia severity. Since functional data about apathetic symptoms in neurodegeneration is still lacking, the present study aimed to investigate large-scale brain networks involved in apathy syndrome in patients with FTD and PD. Functional neuroimaging methods were employed in order to undertake a more integrative and quantitative analysis of the possible neural association of apathy in neurodegeneration.

## 2. Materials and Methods

The study sample includes a total of 60 subjects divided into 20 apathy patients: 7 with primary diagnosis of FTD (mean age 64.6 ± 7.9) and 13 with primary diagnosis of PD (mean age 67.0 ± 7.3); 20 non apathy patients: 7 FTD (mean age 68.3 ± 7.2) and 13 PD (mean age 67.7 ± 8.2); and then 20 healthy controls (HC) matched for age and sex. The three groups participated in a research protocol conducted at the IRCCS SDN that included a clinical evaluation of apathy and a 3T magnetic resonance imaging (MRI) protocol. All participants were recruited if they met the following criteria: (i) lack of current or past history of alcohol or drug abuse, (ii) lack of current or past history of major psychiatric illnesses, and (iii) lack of current or past use of psychoactive medications. All patients were assessed by an expert neurologist and psychologist. Following this, patients with incidental brain focal lesion in MRI examination or excessive vascular load were excluded.

The apathy evaluation scale (AES) was used to investigate and measure the presence of apathy in patients with FTD [13] (range 1–72, cutoff value 38). The apathy scale [14] has been used to investigate and measure the presence of apathy in patients with PD [3] (range 1–42, cutoff value 14).

Each participant provided written informed consent approved by the local Ethics Committee of IRCCS Pascale and according to the ethical standards laid down in the 1964 Helsinki Declaration and its later amendments. 

MRI was acquired on a Biograph mMR 3T scanner (Siemens Healthcare, Erlangen, Germany). A 12-channel head coil was used in a customized neurological protocol (total scan time 55 min) including the following structural and functional sequences:3D T1-Magnetization Prepared Rapid Acquisition Gradient Echo (MPRAGE), voxel size 0.8 × 0.8 × 0.8 mm^3^, Field of View (FOV) 214 mm × 214 mm^2^, TR/TE/TI = 2400/2.25/1000 ms, scan time 5:03.Resting-state fMRI, sequence Echo Planar Imaging-Gradient Echo (EPI-GRE), voxel-size 4 × 4 × 4 mm^3^, TR/TE = 1000/21.4 ms, 350 measurements, bandwidth: 2230 Hz, scan time 6:02.

For structural image processing, the parcellations of morphological T1 weighted 3D images of HC, and the PD and FTD groups were processed with the FreeSurfer v5.1 toolkit (Charlestown, MA) [15]. Briefly, this processing includes spatial inhomogeneity correction, non-linear noise reduction, skull-stripping, subcortical segmentation, intensity normalization, surface generation, topology correction, surface inflation, registration to a spherical atlas, and cortical thickness calculation [16]. Consequently, the result was normalized by the ratio with the estimated total intracranial volume (eTIV). Then, a two-tailed two-sample *t*-test corrected for Bonferroni multiple comparisons (significant *p*-value < 0.0004) was performed to compare brain morphological parameters (cortical volume and cortical thickness) between groups.

Regarding functional image processing, fMRI data were analyzed with functional connectivity toolbox v19.c (CONN, Cambridge, MA) [17] and SPM v12 software (Statistical Parametric Mapping: The Analysis of Functional Brain Images, London, UK). Preprocessing was carried out in CONN using a pipeline that includes realignment, slice-timing, normalization in the Montreal Neurological Institute (MNI) space of functional images, outlier detection with ART-based scrubbing, smoothing, and physiological denoising [17].

We conducted a first-level statistical analysis to assess subjects’ resting-state brain activations. Then, we devised a second-level data analysis to assess differences in functional connectivity (FC) between the three groups. First, we evaluated FC differences between apathetic patients, HC, and non-apathetic patients by performing a CONN-based, seed-to-seed analysis. Then, we performed two seed-to-seed analyses (i.e., FTD vs. HC and PD vs. HC) to investigate whether FC differences were due to the apathy as trait or as belonging to a primary pathological condition. Finally, to validate the FC differences in apathetic patients, the same analysis was carried out between non-apathetic patients (FTD-PD groups) and HC. A *p*-value of 0.05 corrected for false-discovery rate (FDR) multiple comparison [18] was considered significant for FC analysis.

## 3. Results

Structural analysis did not show significant differences in brain parcels volumes and cortical thickness between apathetic patients and HC and between non-apathetic patients and HC. Regarding functional seed-to-seed analysis, resting-state paradigm showed a significant difference between apathetic patients (both FTD and PD) and HC and also a significant difference between apathetic patients and non-apathetic patients (both FTD and PD), highlighting a hypoconnectivity between the left planum polare as seed and right pre- and post-central gyrus (*p* = 0.002 and *p* = 0.004, respectively) considered as targets. T-score and *p*-value FDR corrected in these seeds are resumed in Table 1 and Figure 1. This result was confirmed also when the primary diagnosis was revealed in the apathy group in seed-to-seed analysis between apathetic FTD patients and HC (pre- and post-central gyrus *p* = 0.015 and *p* = 0.041, respectively) and between apathetic PD patients and HC (pre- and post-central gyrus *p* = 0.016 and *p* = 0.021, respectively). T-score and *p*-value FDR corrected in these seeds are resumed in Table 2. This pattern across apathetic patients was not confirmed when non-apathetic FTD-PD patients vs HC was considered. Global results from functional seed-to-seed analysis between apathetic patients and HC and between non-apathetic patients and HC are resumed in Appendix A, respectively. 

## 4. Discussion

Apathy is one of the most widespread behavioral symptoms associated with major neurodegenerative diseases, such as AD, PD, and FTD. In experimental design, we included PD and FTD patients who had either apathetic and non-apathetic traits and compared them with HC. Considered that apathy is the most common condition among behavioral disorders in AD, we decide to not include AD patients in the analysis due to the fact that it is less common to find non-apathetic AD patients to disentangle apathy tract from the ongoing neurodegenerative process [8,19]. In the first analysis, FC was evaluated among apathetic patients (i.e., both PD and FTD) and HC. Subsequently, FCs in apathetic PD vs. HC and apathetic FTD vs. HC were evaluated to find common patterns eventually involved in apathy. Finally, to investigate whether such neural alterations were due to the underlying neurodegenerative pathology, we replicated the analysis by considering two independent patients’ samples (i.e., non-apathetic PD and FTD). In these groups, functional differences were no longer detected. 

Results highlighted an FC reduction between the left PP and both the right precentral-postcentral gyrus in apathetic FTD and PD patients as compared to HC. PP, as a temporal region located in the auditory cortex anteriorly, is implicated in external-auditory-stimuli processing, particularly in speech and pitch processing, and in emotional processing [20]. Indeed, PP appears to be involved also in the elaboration of emotion-related sounds. In particular, FC between the left PP and the ventral striatum increases during threat-music-stimuli processing, probably due to the survival instinct [21]. This is supported by increased FC between auditory and sensorimotor cortical regions (i.e., inferior parietal lobe, supplementary motor area, and premotor cortex) while perceiving fear-related auditory stimuli, probably due to a fight-or-flight response. The authors also showed strong FC between the auditory cortex (i.e., PP and planum temporale) and limbic and paralimbic structures, such as insula, striatum, cingulate, orbitofrontal cortex, and neocortical areas [21]. These findings may indicate the involvement of the PP area in emotional processing. Furthermore, the interconnection we found between PP and motor areas might be implicated in external stimuli’s motor response. The alteration of this network in both apathetic FTD and PD patients (i.e., the hypoconnectivity between PP and pre-post central gyrus) might be associated with a decrease in external-stimuli perception priority, leading to a loss in motor response. One may interpret these results in terms of specific apathy-related alterations, namely, as simultaneous reduction of emotion/external stimuli processing. Several studies demonstrated how different network alterations may justify apathetic patients’ inability to associate rewards with their behavior (i.e., auto-activation) [1]. For instance, PD patients’ auto-activation inability seems to share the same alterations subtending their motor symptoms. Indeed, these processes might be regulated by the neuroanatomical circuits, such as the motor cortex, premotor cortex, and supplementary motor areas [22]. Consistently, we found FC alterations between pre-post central gyrus and PP. Additionally, in FTD patients, the possible prefrontal and temporolimbic network alteration seems to explain apathy symptoms. Specifically, prefrontal structure plays an important role in executive functions and motivation processing; instead, temporolimbic structures seem to be involved in reward and emotional processing [23]. According to Eslinger et al., apathy symptoms in FTD results correlated with atrophy of basal ganglia and specifically to the head of the caudate, including the ventral striatum, which is implicated in emotional, cognitive, and motor aspects of motivational behavior [6]. These results supported the theory that behavioral compromising in bvFTD (specifically apathy symptoms) was related to the cohesion of social, emotional, and execution impairment that depends on several functional and anatomical damage of the prefrontal and temporolimbic network. An important finding of our study is represented by the FC-independent alteration from structural impairment. Indeed, our results showed no significant alteration of cortical thickness in apathetic PD and FTD groups compared to HC. This result, supported by different studies focused on structural MRI assessment in apathy, reflects the putative role of FC alteration in the development of apathy symptoms. Additionally, our results showed left lateralization in FC. According to Baggio et al., this lateralization especially in PD patients may correlate with the highest probability of developing apathy syndrome [24].

In conclusion, our results show a lower FC between PP and pre- and postcentral gyrus as a common pattern in FTD and PD apathetic patients compared to HC. This alteration may subtend the involvement of neural pathways implicated in a specific reduction of information/elaboration processing and motor outcome in apathetic patients. Moreover, our study indicated the putative role of temporal structures during the elaboration of emotional-external stimuli. Indeed, the interconnection between PP and motor-sensorimotor cerebral areas involved in motor response evocation seems to be commonly impaired in apathetic patients.

## Figures and Tables

**Figure 1 jpm-11-00679-f001:**
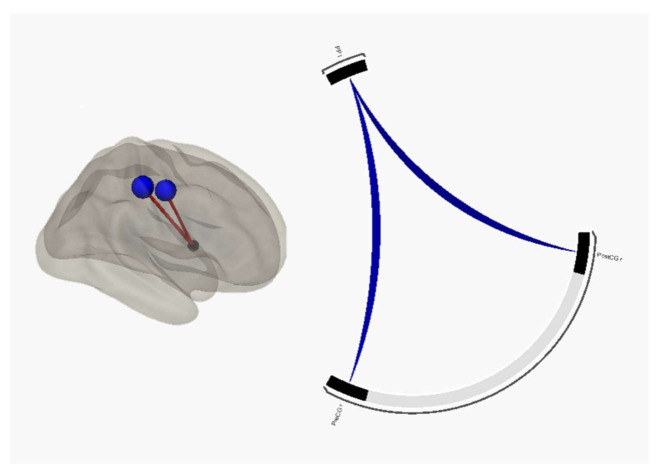
fMRI seed-to-seed 3D representation and connectogram showing the hypoconnectivity between left planum polare and right pre- and post-central gyrus.

**Table 1 jpm-11-00679-t001:** fMRI seed-to-seed results from resting-state paradigm with functional connectivity between apathetic patients and HC group and between apathetic patients and non-apathetic patients group (higher connectivity between seeds have a positive value of T-score, and lower connectivity have a negative value of T-score; p-FDR: *p*-value corrected for false discovery rate).

**Apathetic patients > HC**			
**Seed**	**Targets**	**T-score**	**p-FDR**
Planum polare left	Postcentral gyrus right	−5	0.002
	Precentral gyrus right	−4.7	0.004
**Apathetic patients > Non-apathetic patients**			
**Seed**	**Targets**	**T-score**	**p-FDR**
Planum polare left	Postcentral gyrus right	−3.5	0.041
	Precentral gyrus right	−3.6	0.041

**Table 2 jpm-11-00679-t002:** fMRI seed-to-seed results from resting-state paradigm with functional connectivity between single apathetic FTD and PD group and HC groups (higher connectivity between seeds have a positive value of T-score, lower connectivity have a negative value of T-score. p-FDR: *p*-value corrected for false discovery rate).

**Apathetic FTD > HC**			
**Seed**	**Targets**	**T-score**	**p-FDR**
Planum polare left	Precentral gyrus right	−4.5	0.015
	Postcentral gyrus right	−4	0.041
**Apathetic PD > HC**			
**Seed**	**Targets**	**T-score**	**p-FDR**
Planum polare left	Postcentral gyrus right	−4.2	0.016
	Precentral gyrus right	−3.7	0.021

## Data Availability

The datasets generated for this study are available on request to the corresponding author.

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
