# Peer review of "Identifying a Common Functional Framework for Apathy Large-Scale Brain Network"

_jpm, 2021, doi:10.3390/jpm11070679_

Round 1

Reviewer 1 Report

This study investigated neurological correlates of apathy. Apathy is an intriguing condition not only in clinical care but also in neurosciene itself. Thus, the topic is desirable. I believe the manuscript can be improved from multiple aspects as follows.

Major concerns:
The author compared apathetic patients with HC and non-apathetic patients with HC to confirm the observed differences are not due to the underlying neurodegenerative diseases. Why were the comparisons between apathetic and non-pathetical patients not performed?

The rsfMRI preprocessing is incomplete. The ART motion scrubbing, physiology noise removing, and smoothing were not performed. The author may want to read CONN manuals for details.

What were brain morphological parameters compared? 

Author Response

Reviewer 1:

This study investigated neurological correlates of apathy. Apathy is an intriguing condition not only in clinical care but also in neuroscience itself. Thus, the topic is desirable. I believe the manuscript can be improved from multiple aspects as follows.

Major concerns:
The author compared apathetic patients with HC and non-apathetic patients with HC to confirm the observed differences are not due to the underlying neurodegenerative diseases. Why were the comparisons between apathetic and non-pathetical patients not performed?

Response: We thank the Reviewer for the positive comment and careful review. We performed the functional connectivity analysis between apathetic patients (PD and FTD) and non-apathetic patients. The results are in line with those described in the manuscript (hypoconnectivity between left planum polare as seed and pre-and-postcentral gyrus considered as targets). Those additional results have been added in the manuscript and in table 1.

The rsfMRI preprocessing is incomplete. The ART motion scrubbing, physiology noise removing, and smoothing were not performed. The author may want to read CONN manuals for details.

Response: We thank the Reviewer for the positive comment, the abovementioned preprocessing steps were performed in the CONN preprocessing pipeline. We added those details in the manuscript.

What were brain morphological parameters compared?

Response: We thank the Reviewer for the comment, we extracted cortical volume and cortical thickness from freesurfer brain parcels, and then we compared those features between groups. The manuscript was edited accordingly.

Reviewer 2 Report

Alfano et al have shown a significant hypoconnectivity between apathetic patients of FTD and PD from control, while no functional differences were observed in non-apathetic patients from the same disease populations. The study design is strong and findings are promising too. However, I have the following suggestions for this manuscript.

  1. The addition of a table for the volume of the brain, cortical thickness and, intracranial volume from the control and disease population would provide a reference for future studies, as more studies are targeting to understand a behavior pattern of disease/disorders.
  2. In the introduction section please briefly specify metabolism changes in the frontal cortex, insula, and cerebellum?
  3. Please briefly mention, why the apathy scale is different in two disease populations in the Material and Method section?
  4. Please also mention. The individual scan time of anatomical scans and functions scans and overall total scanning time of the whole MRI protocol.
  5. In the Material and Method section “registration to a spherical atlas and thickness calculation”, should be cortical-thickness calculation.

Author Response

Reviewer 2:

Alfano et al have shown a significant hypoconnectivity between apathetic patients of FTD and PD from control, while no functional differences were observed in non-apathetic patients from the same disease populations. The study design is strong and findings are promising too. However, I have the following suggestions for this manuscript.

  1. The addition of a table for the volume of the brain, cortical thickness and, intracranial volume from the control and disease population would provide a reference for future studies, as more studies are targeting to understand a behavior pattern of disease/disorders.

Response: We thank the Reviewer for the positive comment and careful review. As the structural results were not statistically significant, we didn’t include them in the manuscript with a dedicate table. For the readability of the manuscript we can’t provide every single cortical volume and thickness for each of 130 cortical and subcortical parcels extracted from freesurfer for all the groups. However, the freesufer output will be provided upon reasonable request.

2. In the introduction section please briefly specify metabolism changes in the frontal cortex, insula, and cerebellum?

Response: We thank the Reviewer for the comment, the introduction section has been edited accordingly (reduced metabolism)

3. Please briefly mention, why the apathy scale is different in two disease populations in the Material and Method section?

Response: We thank the Reviewer for the positive comment, apathy scale is different in the two disease population because with the Parkinson disease group there is a dedicated Apathy Scale (Starkstein et al. 1992), which is an abridged version of the apathy scale designed by Marin et al. 1991. While for the FTD groups we used the standard apathy evaluation scale (Raimo et al. 2014).

Starkstein, S.E.; Mayberg, H.S.; Preziosi, T.J.; Andrezejewski, P.; Leiguarda, R.; Robinson, R.G. Reliability, validity, and clinical correlates of apathy in Parkinson's disease. J Neuropsychiatry Clin Neurosci. 1992 Spring;4(2):134-9.

Marin, R.S.; Biedrzycki, R.C.; Firinciogullari, S. Reliability and validity of the Apathy Evaluation Scale. Psychiatry Res. 1991 Aug;38(2):143-62.

Raimo, S.; Trojano, L.; Spitaleri, D.; Petretta, V.; Grossi, D.; Santangelo, G. Apathy in multiple sclerosis: a validation study of the apathy evaluation scale. J Neurol Sci. 2014 Dec 15;347(1-2):295-300.

4. Please also mention. The individual scan time of anatomical scans and functions scans and overall total scanning time of the whole MRI protocol.

Response: We thank the Reviewer for the comment, the individual scan time of anatomical and functional scans were added in the manuscript together with scanning time of the whole MRI protocol.

5. In the Material and Method section “registration to a spherical atlas and thickness calculation”, should be cortical-thickness calculation.

Response: We thank the Reviewer for the comment, “cortical thickness calculation” has been added to the manuscript.

Round 2

Reviewer 1 Report

The authors have addressed my comments properly.